# MMPs and TIMPs Expression Levels in the Periodontal Ligament during Orthodontic Tooth Movement: A Systematic Review of In Vitro and In Vivo Studies

**DOI:** 10.3390/ijms22136967

**Published:** 2021-06-28

**Authors:** Christian Behm, Michael Nemec, Fabian Weissinger, Marco Aoqi Rausch, Oleh Andrukhov, Erwin Jonke

**Affiliations:** 1Clinical Division of Orthodontics, University Clinic of Dentistry, Sensengasse 2A, 1090 Vienna, Austria; christian.behm@meduniwien.ac.at (C.B.); michael.nemec@meduniwien.ac.at (M.N.); marco.rausch@meduniwien.ac.at (M.A.R.); erwin.jonke@meduniwien.ac.at (E.J.); 2Competence Centre for Periodontal Research, University Clinic of Dentistry, Sensengasse 2A, 1090 Vienna, Austria; fabianweissinger95@gmx.at

**Keywords:** orthodontic tooth movement, matrix metalloproteinases, tissue inhibitor of matrix metalloproteinases, periodontal ligament

## Abstract

**Background:** During orthodontic tooth movement (OTM), applied orthodontic forces cause an extensive remodeling of the extracellular matrix (ECM) in the periodontal ligament (PDL). This is mainly orchestrated by different types of matrix metalloproteinases (MMPs) and their tissue inhibitors of matrix metalloproteinases (TIMPs), which are both secreted by periodontal ligament (PDL) fibroblasts. Multiple in vitro and in vivo studies already investigated the influence of applied orthodontic forces on the expression of MMPs and TIMPs. The aim of this systematic review was to explore the expression levels of MMPs and TIMPs during OTM and the influence of specific orthodontic force-related parameters. **Methods:** Electronic article search was performed on PubMed and Web of Science until 31 January 2021. Screenings of titles, abstracts and full texts were performed according to PRISMA, whereas eligibility criteria were defined for in vitro and in vivo studies, respectively, according to the PICO schema. Risk of bias assessment for in vitro studies was verified by specific methodological and reporting criteria. For in vivo studies, risk of bias assessment was adapted from the Joanna Briggs Institute Critical Appraisal Checklist for analytical cross-sectional study. **Results:** Electronic article search identified 3266 records, from which 28 in vitro and 12 in vivo studies were included. The studies showed that orthodontic forces mainly caused increased MMPs and TIMPs expression levels, whereas the exact effect may depend on various intervention and sample parameters and subject characteristics. **Conclusion:** This systematic review revealed that orthodontic forces induce a significant effect on MMPs and TIMPs in the PDL. This connection may contribute to the controlled depletion and formation of the PDLs’ ECM at the compression and tension site, respectively, and finally to the highly regulated OTM.

## 1. Introduction

The periodontal ligament (PDL) is a highly specialized connective tissue, including a heterogenous cell population (PDL cells) [1] and a fibrous extracellular matrix (ECM) [2]. The PDL is essential in sensing and transmitting mechanical forces from the teeth to the alveolar bone, such as during orthodontic tooth movement (OTM) [3]. In addition to bone remodeling, orthodontic tension as well as compression forces cause a continuous re-organization of the PDL’s ECM. This ECM remodeling is mainly achieved by PDL cells [3,4,5,6,7], which contribute to ECM deposition by secreting matrix proteins [8,9], but the remodeling is also related to ECM protein degradation by the expression of various proteolytic enzymes, such as matrix-metalloproteinases (MMPs) [3,5,10].

MMPs are a growing family of calcium-dependent and zinc-dependent endopeptidases, encompassing at least 23 different types which are expressed in human tissues. The various MMP types are divided into six main groups based on the arrangement of their structural domains and their substrate preferences: collagenases (MMP-1, -8, -13 and -18), gelatinases (MMP-2 and -9), stromelysins (MMP-3 and -10), matrilysins (MMP-7 and -26), membrane-type MMPs (MMP-14, -15, -16, -17, -24 and -25) and others (MMP-12, -19, -20, -22 and -27). In addition, several other MMP types exist, which are distinguished from typical MMPs by their unique structural features. All these MMPs are released as inactive zymogen (pro-MMPs), which have to be proteolytically processed for activation [11]. For sustaining homeostatic ECM conditions, MMPs are regulated via their expression, the processing of their zymogens to active MMPs and via endogenous tissue inhibitors of matrix metalloproteinases (TIMPs) [11], which are also expressed by PDL cells [12,13,14,15]. This family of MMPs inhibitors comprises four different types (TIMP-1, -2, -3 and -4), which bind MMPs unspecifically in a 1:1 stoichiometry. Alterations on the MMPs or TIMPs levels change the MMP/TIMP ratio, resulting in a particular net MMP activity [11].

In the last two decades multiple human studies have investigated the MMPs and TIMPs expression levels during OTM in vitro [3,5,9,12,13,14,15,16,17,18,19,20,21,22,23,24,25,26,27,28,29,30,31,32,33,34,35,36] and in vivo [10,37,38,39,40,41,42,43,44,45,46,47]. All these experimental studies investigated expression levels of single or various combinations of MMPs/TIMPs types under different in vitro (force type, magnitude, application mode and force duration) [3,5,9,12,13,14,15,16,17,18,19,20,21,22,23,24,25,26,27,28,29,30,31,32,33,34,35,36,48] and in vivo (type of appliance, investigated teeth, force magnitude and observation time) [10,37,38,39,40,41,42,43,44,45,46,47] conditions. However, the differences in MMPs and TIMPs expression between tension and compression areas in the PDL and between different orthodontic appliances have never been systematically documented. Hence, the main aim of this systematic review is to review the literature on MMPs/TIMPs and OTM, with a focus on the changes in expression levels of certain MMPs or TIMPs types in the PDL and on possible associations between specific orthodontic force-related parameters (e.g., tension/compression and cyclic/static). In order to achieve this aim, PubMed and Web of Science were systematically scanned for in vitro and in vivo studies with a focus on the expression levels of MMPs and TIMPs in the human PDL during the application of orthodontic forces. For a clearer view, in vitro and in vivo studies were separately analysed with respect to the changes in MMPs and TIMPs expression levels during the application of mechanical forces or during orthodontic tooth movement, respectively. In the discussion section, results from the in vitro and in vivo part were merged and interpreted together.

## 2. Results

### 2.1. Systematic Search Results

The systematic search and eligibility process, which is presented as PRISMA Flow Diagram in Figure 1, identified 28 in vitro [3,5,9,12,13,14,15,16,17,18,19,20,21,22,23,24,25,26,27,28,29,30,31,32,33,34,35,36] and 12 in vivo [10,37,38,39,40,41,42,43,44,45,46,47] studies, which were included in this systematic review. Overall, 9 in vitro studies were excluded: one study due to the use of an unclear cell type [49]; one study due to the use of cells isolated from tissues other than the PDL [50]; one study due to the use of a non-human cell line [51]; four studies due to the use of 3D cell culture models [41,48,52,53]; one study which did not use force application [54]; and one study due the use of cells isolated from the gingiva in a 3D cell culture model [55]. Concerning in vivo studies, 6 articles were excluded: three studies did not use untreated healthy patients as control [56,57,58]; one study which investigated expression levels in the PDL tissue [59]; one study which used only periodontitis subjects [60]; and one study due to an unclear outcome [61]. Due to a large heterogeneity of the intervention parameters of the included studies, no quantitative meta-analysis was feasible.

### 2.2. In Vitro Studies

#### 2.2.1. Sample Parameters

The sample parameters of all included in vitro studies are described in Table 1. All 28 included in vitro studies using human cells isolated from the PDL [3,5,9,12,13,14,15,16,17,18,19,20,21,22,23,24,25,26,27,28,29,30,31,32,33,34,35,36]. There were 11 studies that named these cells as “human PDL cells” [3,12,16,22,23,24,27,29,31,32,35], 16 studies as “human PDL fibroblasts (HPdLFs)” [5,9,13,14,15,17,18,19,20,21,25,26,28,30,33,36] and one study as “human periodontal ligament derived mesenchymal stromal cells (hPDL-MSCs)” [34]. Primary PDL cells/fibroblasts were isolated from third molars or premolar in 8 [19,20,21,22,24,31,32,34] or 9 [9,12,16,26,27,29,30,33,36] studies, respectively. The remaining studies did not specify the teeth which were used for cell isolation, in which four studies used commercially available HPdLFs [13,14,17,18] and one study immortalized PDL cell lines [23]. Not all studies that used self-isolated PDL cells/fibroblasts specified the number of included donors and their gender and age. The number of used donors ranged from 1 to 6 patients [3,9,12,15,22,24,25,26,28,32,34,36]. Only two studies used a much higher number of donors: 16 [16] and 36 [19]. Three studies used only female donors [12,15,16], two studies only male [5,25] and three studies had a mixed donor population [19,22,28]. The other studies did not specify the donor’s gender. The donor age ranged from 4.5 years to 40 years [5,12,16,22,24,25,28,31,32,33]. Two studies specified the donors’ age with “young” [9,29]. Residual studies did not specify the donors’ age.

#### 2.2.2. Intervention Parameters

The intervention parameters of the included in vitro studies are summarised in Table 2. There were 17 studies that applied tensile mechanical strain, from which seven [5,15,17,23,28,33,34] used a static application mode and 10 [3,9,12,18,22,24,27,29,30,35] used a cyclic application mode. The frequency of the applied cyclic tensile strain varied between 0.005 and 0.5 Hertz (Hz). One study did not specify the used frequency [18]. All 17 studies applied mechanical tensile strain by stretching flexible-bottomed culture dishes and specifying the force magnitude by the percentage of elongation, which ranged from 1% to 110%. One study indicated the used force magnitude in kilopascal (20 kPa = 2 N/cm^2^) [15]. In most cases, the duration of force application ranged between 0.25 to 48 h. Only one study used an extended period of force application (from one to 7 days) [24]. The 15 studies evaluated MMPs/TIMPs expression levels immediately after force application, whereas two studies analysed expression levels several hours after force application was stopped (0–12 h [5] and 2–48 h [3] after treatment was finished). Nine included studies used compressive mechanical load. In 8 studies, the static application mode was applied [13,14,16,19,20,21,25,26]. Only one study investigated, in addition to the static application form, the cyclic application form, using two different frequencies (30 min or 15 min with 5 min intervals) [36]. Seven studies applied compressive forces by centrifugation [13,16,19,20,21,26,36] and one study by compressive plates [14]. One study only stated that a “static weight application” was used [25]. The used force magnitudes were indicated as g/cm^2^, varying from two to 36.3 g/cm^2^. Three studies specified applied force magnitudes in g (141 g) [19,20,21] and one study in cN/mm^2^ (2–4 cN/mm^2^) [14]. The duration of force application ranged between 10 min to 24 h. Five studies verified MMPs/TIMPs expression levels immediately after force application was stopped [13,14,16,25,26], whereas three studies used a 24 h post-treatment incubation period [20,21,36]. One study extended this period to 72 h [19]. Two studies used a steady laminar shear flow in a static mode [31,32]. The force magnitude varied between 6 to 12 dyn/cm^2^ and the force duration between 4 to 12 h. Expression level evaluation was carried out immediately after force application was completed.

#### 2.2.3. Risk of Bias Assessment

Figure 2 and Appendix A show the methodological and reporting quality of all included in vitro studies. All in vitro studies showed high methodological quality (Figure 2a and Appendix A). An appropriate control group selection, complete outcome data, a controlled exposure, a valid test system and a detailed description of the treatment were found in all 28 studies [3,5,9,12,13,14,15,16,17,18,19,20,21,22,23,24,25,26,27,28,29,30,31,32,33,34,35,36]. No selective outcome reporting was observed in any of these studies. One study did not accurately describe the statistical analysis [20]. The “Conflicts of Interest” statement and funding source were stated in 15 studies [5,12,13,14,17,18,20,21,24,25,28,31,32,34,62]. Sample size determination was not observed in any of the investigated studies [3,5,9,12,13,14,15,16,17,18,19,20,21,22,23,24,25,26,27,28,29,30,31,32,33,34,35,36]. The overall reporting quality of included in vitro studies was sufficient (Figure 2b and Appendix A). The description of the scientific background and of the objectives defined the experimental outcomes and cell maintenance conditions were sufficiently stated in all in vitro studies [3,5,9,12,13,14,15,16,17,18,19,20,21,22,23,24,25,26,27,28,29,30,31,32,33,34,35,36]. The justification for the used model was inaccurately described or missing in five studies [13,20,21,23,35], whereas one study contained a deficient description of the study design [18]. The ethical statement was missing in 7 studies [9,17,22,23,26,31,35]. One study did not mention statistical analysis in the material and methods section [20], whereas none of the included in vitro studies contained a description of measurement precision [3,5,9,12,13,14,15,16,17,18,19,20,21,22,23,24,25,26,27,28,29,30,31,32,33,34,35,36].

#### 2.2.4. In Vitro MMPs Express Levels

Changes in expression levels of all investigated MMPs are summarised in Table 3 and Table 4. In total, 11 different MMP types (MMP-1, -2, -3, -8, -9, -10, -11, -12, -13, -14 and 15) were analysed in 28 in vitro studies by using various screening approaches, such as quantitative polymerase chain reaction (qPCR), enzyme-linked immunosorbent assay (ELISA) and Western blot analysis. Three studies investigated MMPs’ enzymatic activity by zymogram [9,19,20] and one study used an oligo-DNA chip analysis [35]. MMP-1 expression was investigated in 12 studies [3,5,15,21,22,24,26,29,31,34,36,63]. Eight studies identified a significant increase in MMP-1 expression [3,5,15,16,24,26,31,36]. Four studies did not reveal changes in MMP-1 levels [21,22,29,34]. Ten studies investigated MMP-2 expression levels [3,9,15,19,20,29,30,31,32,34]. Four studies observed a significant increase in MMP-2 expression levels [3,9,15,31], whereas two studies observed decreased MMP-2 enzymatic activities [19,32]. Four studies did not show any changes in the expression levels [20,29,30,34]. Lisboa et al. [21], Long et al. [22], Nemoto et al. [24] and Tantilertanant et al. [3] detected MMP-3 expression levels. One study observed an increase [3], one study observed a decrease [24] and two studies reported no changes [21,22] in MMP-3 expression levels. MMP-8 expression levels were examined in nine different studies [3,12,13,14,17,18,27,28,30], in which four studies reported increased [13,14,17,18] and two studies decreased [12,27] expression levels. Three studies did not observe any differences in MMP-8 expression level [3,28,30]. Only one study investigated MMP-9 and reported no changes in its expression level [30]. Lisboa et al. [21], Wescott et al. [30] and Ziegler et al. [33] observed MMP-10 expression levels. One study observed a significant decrease in MMP-10 expression levels [33], whereas two studies did not detect any differences [21,30]. Only one study examined MMP-11, observing a significant decrease in its expression level [27]. Two studies observed a significant increase in MMP-12 expression levels [23,33] and MMP-13 expression levels were shown to be increased in two other studies [25,33]. Tantilertanant et al. [3] revealed significantly enhanced MMP-14 expression levels. MMP-15 expression levels were observed to be increased [35] and decreased [27] in one study, respectively.

#### 2.2.5. In Vitro TIMPs Express Levels

Changes in the expression levels of all investigated MMPs are summarised in Table 4. In total, three different TIMP types (TIMP-1, -2 and -3) were analysed in 28 in vitro studies, using qPCR, ELISA and/or Western blot analysis. There were 15 studies that investigated the TIMP-1 expression level [3,12,13,14,15,17,18,22,23,26,27,29,31,34,36]. Seven studies revealed a significant increase in TIMP-1 expression levels [12,15,17,18,26,29,31], whereas in two studies TIMP-1 expression levels significantly decreased [13,14]. Six studies did not observe any changes in TIMP-1 expression levels [3,22,23,27,34,36]. Ten studies investigated the expression levels of TIMP-2 [3,9,12,15,22,23,26,27,29,31]. Significantly increased TIMP-2 expression levels were observed in four studies [9,15,26,29], whereas one study observed decreased expression [31] and five studies reported no changes in TIMP-2 expression [3,12,22,23,27]. TIMP-3 was observed in three different studies, showing no changes in its expression level [12,23,27].

### 2.3. In Vivo Studies

#### 2.3.1. Patient Parameters

The patient’s parameters of all included in vivo studies are described in Table 5. All 12 in vivo studies included healthy patients without any relevant underlying healthy conditions [10,37,38,39,40,41,42,43,44,45,46,47]. Eight studies included male and female patients [10,38,40,41,42,44,45,46], whereas seven of these studies definitely contained a higher number of female subjects [10,38,40,41,42,44,46]. Only one study comprised a balanced number (six versus six) of female and male patients [45]. Four studies did not mention the patients’ gender [37,39,43,47]. The age of included patients varied from 10 to 45 [10,37,38,39,40,41,42,43,44,45,46,47], in which most of the participating patients were young people and young adults below 25 years [10,38,39,41,43,45]. Six studies mainly used a narrow age range comprised of patients between 10 and 20 years [10,38,39,41,43,45], whereas 5 studies used a broader range between 11 and 45 years [37,40,42,44,47]. One study used two age cohorts including 10 adolescents (13–15 years) and 10 adults (21–39 years), respectively [46]. The sample size varied between 5 to 28 included patients [10,37,38,39,40,41,42,43,44,45,46,47].

#### 2.3.2. Intervention Parameters

The intervention parameters of the included in vivo studies are summarised in Table 6. All 12 in vivo studies measured MMPs/TIMPs expression levels in the gingival crevicular fluid (GCF) [10,37,38,39,40,41,42,43,44,45,46,47]. Since one study used PDL tissue to analyse the expression levels, this study was excluded [59]. The pool of orthodontic treated teeth was consequently very heterogeneous, including six [10,37,38,39,42,43], one [41] and another study [46] using canines, mostly first premolars and incisors, respectively. Four studies used both canines and incisors [40,44,45,47]. Additionally, six studies applied OTM to the teeth in the upper jaw [10,38,41,42,43,46], whereas three studies treated teeth in both the upper and lower jaw [40,44,47]. Three studies did not mention the upper or lower localisation of OTM subjected teeth [37,39,45]. The method used to apply orthodontic forces varied between included studies. The appliance types are described in detail in Table 6. Three studies did not specify the used orthodontic appliance [39,42,45]. The applied orthodontic force magnitude was mentioned in five studies, which specified the force with 150 g [10,42], 100 g [43], 50 centinewton (cN = 51 g) [37] or 150 cN (=153 g) [41]. The duration of applying orthodontic forces ranged between one and five months [37,41,44,45,46]. One study started to collect GCF samples after the completion of OTM. However, this study did not state an exact treatment duration [38]. Six studies did not mention the duration of orthodontic treatment [10,39,40,42,43,47]. The included studies used a wide range of observation times. Eight studies measured MMPs/TIMPs levels before starting the treatment and ranged between seven days to just before applying orthodontic forces [37,38,39,40,42,44,45,46,47]. MMPs/TIMPs expression levels were measured during orthodontic treatment in various intervals. The observation time points ranged from several hours to several months after orthodontic force application. One study observed MMPs/TIMPs expression levels even 12 months after bracket bonding [47]. A huge variability was observed concerning the used control groups. Four studies compared orthodontic treatment affected MMPs/TIMPs expression levels with GCF samples collected from the same teeth immediately to seven days before starting the treatment [37,42,45,47] or they used orthodontic untreated teeth as control [10,38,41,46], respectively. Grant et al. [43] and Zhang et al. [39] used both control types. For orthodontic untreated controls, studies used antagonistic teeth [10], second molars [43], same tooth from the other jaw [46] or contralateral teeth [38,39]. One study did not specify the used control teeth [41]. A non-treated control cohort with systemically healthy subjects was used in two other studies [40,44].

#### 2.3.3. Risk of Bias Assessment

Figure 3 and Table 3 showed the risk of bias assessment of all included in vivo studies. The overall risk of bias assessment of the included in vivo studies was good [10,37,38,39,40,41,42,43,44,45,46,47]. The use of standard criteria for measurement of the conditions and the outcome measurements in a valid and reliable manner were assessed with a lower bias risk for all included in vivo studies [10,37,38,39,40,41,42,43,44,45,46,47]. Two studies did not clearly define the criteria for inclusion [40,44]. Only one study contained a sufficient description of the study subjects and the utilised settings [41]. Co-founding factors were identified in three studies [37,43,47], in which two of these studies stated strategies to deal with these co-founding factors [37,43]. Appropriate statistical analysis was not mentioned in one study [44].

#### 2.3.4. In Vivo MMPs Expression Levels

Changes in expression levels of all investigated MMPs are summarised in Table 7. MMP-1 expression was reported in a total of five studies [10,39,40,41,44]. Three studies observed a significant increase in MMP-1 expression levels [10,39,41]. In two studies MMP-1 expression was not detectable [40,44]. A significant increase in MMP-2 expression levels was observed in a total of three different studies [10,39,41]. Two studies investigated the MMP-3 expression levels [39,42]. One study observed a significant increase in MMP-3 expression levels [39], whereas another study reported significant changes in MMP-3 expression levels depending on the force duration [42]. Six studies investigated the expression levels of MMP-8 [39,40,41,44,45,47]. Three studies reported a significant increase in MMP-8 expression levels [39,40,44]. One study found no differences [47] and in another study MMP-8 was not detectable [41]. Another study demonstrated significant changes in MMP-8 expression levels; however, this depended on the force duration [45]. MMP-9 expression levels were observed in a total of seven studies [37,38,39,41,42,43,57]. Five studies observed a significant increase in MMP-9 expression [37,38,39,41,43]. Capelli et al. observed significant changes in MMP-9 levels with significant increases and decreases between one hour and 80 days [42]. Rody et al. did not detect changes in MMP-9 expression [46]. Bildt et al. [41], Capelli et al. [42] and Zhang et al. [39] investigated MMP-13 expression levels. One study observed a significant increase in MMP-13 expression levels [39], whereas one study did not detect any MMP-13 expression [41]. Capelli et al. also observed significant changes in MMP-13 expression with significant increases and decreases between one hour and 80 days [42]. Only one study investigated MMP-14 and demonstrated a significant increase in its expression level [39].

#### 2.3.5. In Vivo TIMPs Expression Levels

Changes in expression levels of all investigated TIMPs are summarised in Table 8. In total, two different TIMP types (TIMP-1 and TIMP-2) were analysed using Luminex multi-analyte technology or reverse zymogram [41,43]. Two studies investigated TIMP-1, observing a significant increase in its expression level [41,43]. The same two studies also investigated TIMP-2 [41,43]. While Grant et al. found significantly increased TIMP-2 expression levels [43], Bildt et al. did not detect any positive TIMP-2 signals [41].

## 3. Discussion

The PDL undergoes a constant physiological turnover, which is partly executed by the ECM protein degrading MMPs and their local inhibitors TIMPs [59]. Since this ECM remodelling is affected by applied orthodontic forces [3,4,5,6,7], a plurality of in vitro [3,5,9,12,13,14,15,16,17,18,19,20,21,22,23,24,25,26,27,28,29,30,31,32,33,34,35,36] studies have already investigated the impact of orthodontic forces on MMPs and TIMPs expression levels in PDL cells. The importance of changes in MMPs and TIMPs expression levels in the PDL during OTM was verified by various in vivo studies [10,37,38,39,40,41,42,43,44,45,46,47] and also by several animal studies, which reported decreased OTM after the inhibition of MMPs with synthetic MMPs inhibitors [64,65]. However, all of these studies show a huge variability in their outcomes and, hence, this systematic review aimed to analyse the literature on MMPs and TIMPs during OTM and examine their potential association with specific orthodontic force-related parameters. This may result in a clearer understanding on the potential differences in MMPs and TIMPs expression levels between compression and tension areas of the PDL and between the usage of different orthodontic appliances during orthodontic treatment. It may cause an overall clearer picture on the role of the different MMPs and TIMPs types on PDL remodelling during OTM.

Most of the included in vitro studies [3,5,9,12,13,14,15,16,17,18,19,20,21,22,23,24,25,26,27,28,29,30,31,32,33,34,35,36] applied simulated orthodontic forces to cells isolated from the human PDL, which are known to be mechano-sensitive. These studies investigated a huge number of different MMPs and TIMPs, verifying a significant influence of mechanical forces on MMPs and TIMPs expression levels in PDL cells. Since the ECM of the PDL consist mainly of type I and III collagens [9], these in vitro studies mainly focused on the expression levels of the collagenases MMP-1 and MMP-8. Additionally, the expression of the gelatinase MMP-2 was often investigated [3,5,9,12,13,14,15,16,17,18,19,20,21,22,23,24,25,26,27,28,29,30,31,32,33,34,35,36]. The expression levels of these three and also of the other MMPs were increased by orthodontic forces. This increasing effect was observed by various combinations of the used treatment parameters: By using tensile or compressive forces in combination with static or cyclic application mode at high and low force magnitudes and during a broad treatment period (30 min to 7 days). However, certain studies also observed no impact [3,20,21,22,28,29,30,34] or even a decrease [12,19,24,27,32,33] in the expression levels of specific MMPs. The heterogeneity in results may be explained by the use of different combinations of force type, application mode, force magnitude and duration. Additionally, the inconsistency in the used teeth for cell isolation in donors’ gender and age may also contribute to this variable influence. In order to overcome this inter-study heterogeneity, future studies in this field should test and directly compare MMPs expression levels under different combinations of force type, application mode, force magnitude and duration. The usage of cells from the same passage and donor will reduce donor variability.

Several in vitro studies [3,15,21,22,24,27,29,30,31,33,34] determined the expression levels of multiple MMPs in parallel. The applied mechanical forces caused mainly similar effects on the expression levels of different MMPs within one study. This indicates that mechanical forces with specific parameters regulate different MMP types in PDL cells via presumably the same methods, e.g., their expression, the processing of their zymogens to active MMPs or influencing TIMPs [11]. However, Nemoto et al. [24] observed increased MMP-1 and decreased MMP-2 expression levels. Tantilertanant et al. [3] showed no effect of applied mechanical forces on MMP-8 expression, whereas MMP-1, MMP-2 and MMP-3 expression levels were significantly increased.

Since TIMPs are important endogenous inhibitors of MMPs [11], it is not surprisingly, that mechanical forces also affect TIMPs expression levels in PDL cells, which was proven in several in vitro studies [9,12,13,14,15,17,18,26,29,31]. Multiple studies, using mainly tensile forces, observed a significant increase in the expression levels of TIMP-1 and TIMP-2 [12,15,17,18,29,31]. Two studies revealed a decrease in TIMP-1 expression when applying compressive mechanical forces between 3 and 12 h [13,14]. In contrast, Redlich et al. [26] revealed a significant increase in TIMP-1 and TIMP-2 expression levels, however, after applying compressive mechanical forces only for 10 to 60 min. This indicates that upregulated TIMP levels in the tension area may be essential for inhibiting MMPs to stop ECM degradation, which may indirectly contribute to new ECM formation. At the compression site, a delayed decrease in TIMPs expression levels may favour MMPs’ enzymatic activity and consequently ECM degradation.

The results of in vitro studies are mainly supported by the outcome of included in vivo studies. All in vivo studies [10,37,38,39,40,41,42,43,44,45,46,47] used GCF samples from healthy human orthodontic patients to investigate the influence of orthodontic forces on the expression levels of a broad range of MMPs, including MMP-1, MMP-8 and MMP-13 collagenases. These studies showed a high variability in investigated teeth, used appliance types, used force durations and used controls. Nevertheless, the expression levels of MMPs were mainly increased by applying orthodontic forces within a broad time range after orthodontic treatment initiation at both the compression and tension areas. The increased MMPs’ expression levels were partially higher at the compression zone. These results are in accordance with Garlet et al. [59], who showed significantly increased MMP-1 expression levels in the PDL tissue at both the tension and compression areas with a significant higher expression level at the compression zone. This indicates a potential higher importance of MMPs-driven ECM protein degradation at the compression site. Different appliance forms are known to cause different types of forces. In our systematic review, three studies used nickel-titanium coil springs [10,37,41], which results in static forces during OTM [66,67]. Four studies used multibracket appliances [38,44,46,47], which causeed, together with occlusal forces, a cyclic orthodontic load [17,68]. Our systematic review revealed no definitive differences in MMPs expression levels concerning the two different force application forms in vivo [10,37,38,39,40,41,42,43,44,45,46,47]. This conclusion is also supported by in vitro studies [3,5,9,12,13,14,15,16,17,18,19,20,21,22,23,24,25,26,27,28,29,30,31,32,33,34,35,36]. Only one in vivo study demonstrated a significant decrease in MMP-8 expression levels. However, this was observed only after 1 week of orthodontic treatment and was followed by a significant increase in the expression level [45]. Rody et al. [46] and Shirozaki et al. [47] showed no significant influence of orthodontic treatment on MMP-8 or MMP-9 expression levels, respectively. It should be noted that, Rody et al. [46] used orthodontic treated teeth from the upper arch and untreated control teeth from the lower jaw. Since orthodontic treatment causes differences in OTM between the upper and lower jaw, this comparison may cause a bias [69].

Only two included in vivo studies investigated TIMP-1 and TIMP-2 expression levels, showing a significant increase at both the tension and compression area up to 4 weeks after beginning the orthodontic treatment [41,43]. Directly in the PDL tissue, Garlet et al. [59] showed a significant increase in TIMP-1 expression at the tension site but no changes in the compression area.

Taken together, it could be possible that mechanical forces mainly upregulate the expression of MMPs in the PDL directly at both the tension and compression site, however, with higher local MMPs concentrations at the compression site. This may be essential for the required depletion of ECM proteins at the compression site. A potential decrease in TIMPs’ concentration by orthodontic compressive forces might further facilitate this process. In contrast, the necessary downregulation of MMPs activity at the tension area may occur predominantly via increased TIMPs’ production and not via the inhibition of MMPs’ gene expression. This may contribute to the essential formation of the PDL’s ECM in the tension area.

The outcome and assumptions of this systematic review have to be handled with caution, since the included in vitro and in vivo studies have several limitations. TIMP-1 and TIMP-2 were analysed only in two in vivo studies [41,43]. Only a few in vitro and in vivo studies have discriminated between active and latent forms of MMP types [41]. All in vitro studies [3,5,9,12,13,14,15,16,17,18,19,20,21,22,23,24,25,26,27,28,29,30,31,32,33,34,35,36] applied orthodontic forces on cells isolated from the human PDL. However, only three studies verified the cell type [22,25,32] and only one study checked the mesenchymal stromal cell character of isolated cells [34]. All included in vivo studies [10,37,38,39,40,41,42,43,44,45,46,47] measured MMPs and TIMPs expression levels in the GCF, which does not necessarily reflect their levels in the PDL. Only one excluded study determined expression levels directly in the PDL [59]. Hence further in vivo studies should investigate MMPs and TIMPs expression levels and activities directly in the PDL. Since applied orthodontic forces markedly differ between the upper and lower jaw [69], there is a bias by comparing the included in vivo studies that used various treated teeth in the upper and lower jaw for sample taking. One study even used tested and control teeth from the upper and lower jaw, respectively [46]. Additionally, different controls (untreated teeth from the same patient versus untreated before OTM) were used in in vivo studies. Hence, future studies should precisely describe their control groups and substantiate the choice of their control type. Due to these inconsistencies, performing a meta-analysis was not possible. Lastly, the risk of bias in in vivo studies could have been reduced by consistently identifying confounding factors and possible strategies to deal with those factors and by describing the study subjects and settings in more detail. This systematic review is further limited by the inclusion of only English written papers and the exclusion of grey literature, such as conference abstracts and dissertations.

## 4. Materials and Methods

This systematic review was conducted in accordance to the Preferred Reporting Items for Systematic Reviews and Meta-Analysis (PRISMA) [70]. Due to the in vitro and in vivo characteristics of included studies, this systematic review was not registered in the PROSPERO database. The whole protocol was independently conducted by two different researchers. The results were compared and discrepancies were discussed until differences were resolved.

### 4.1. Database Search and Screening Strategy

An electronic article search was performed in the PubMed and Web of Science databases. All articles which were indexed in PubMed and Web of Science until 31 January 2021 were included in this review. The search strategy, including only MeSH terms, was created specifically for each database. Pubmed: (“Tooth Movement” OR tooth movement* OR “Tooth Migration” OR tooth migration OR tooth drift* OR tooth displacement OR “Tooth Mobility” OR tooth mobility OR tooth mobilities OR “Orthodontics” OR orthodontic* OR Mechanical force OR orthodontic force) AND (MMP OR MMPs OR Metalloproteinases OR TIMP OR TIMPs OR Tissue inhibitor of metalloproteinase OR Tissue inhibitors of metalloproteinases OR Collagenases OR Gelatinases OR Stromelysins OR Matrilysins OR Collagenase OR Gelatinase OR Stromelysin OR Matrilysin OR Membrane type MMP OR PDL OR parodontal tissue ligament). Web of Science: TS = ((“Tooth Movement” OR tooth movement* OR “Tooth Migration” OR tooth migration OR tooth drift* OR tooth displacement OR “Tooth Mobility” OR tooth mobility OR tooth mobilities OR “Orthodontics” OR orthodontic* OR Mechanical force OR orthodontic force) AND (MMP OR MMPs OR Metalloproteinases OR TIMP OR TIMPs OR Tissue inhibitor of metalloproteinase OR Tissue inhibitors of metalloproteinases OR Collagenases OR Gelatinases OR Stromelysins OR Matrilysins OR Collagenase OR Gelatinase OR Stromelysin OR Matrilysin OR Membrane type MMP OR PDL OR parodontal tissue ligament)).

All found studies were imported into the Mendeley reference manager (Elsevier, The Netherlands) and screened on the basis of the title and abstract. Studies that dealt with the influence of mechanical forces on MMPs and TIMPs in the PDL in vitro and in vivo were included. In a second step, full-texts of all included studies were screened for eligibility on the basis of defined inclusion and exclusion criteria. Additionally, the reference lists of all included studies were manually screened for further relevant literature, which were not recorded during the initial database search. These studies were only included when they met the eligibility criteria. All included studies were divided into in vitro and in vivo for separate qualitative examination. Due to a large heterogeneity within the included in vitro studies concerning the applied force type, mode, magnitude and duration and within included in vivo studies concerning sampling point, appliance type, duration of treatment and observation time, no quantitative meta-analysis was possible.

### 4.2. Eligibility Criteria

Inclusion and exclusion criteria were defined separately for in vitro and in vivo studies and followed the PICO schema. Inclusion criteria for in vitro studies were specified as follows: (P) human PDL cells in a conventional 2D cell culture; (I) in vitro static or cyclic mechanical load; (C) human PDL cells not exposed to mechanical load; (O) evaluation of expression levels and/or enzymatic activities of MMPs and/or TIMPs. In vitro studies were excluded if they fulfilled one of the following exclusion criteria: using 3D cell culture models; usage of non-human PDL cells and/or PDL cells isolated from inflamed PDL tissue (e.g., periodontitis); describing expression of MMPs and/or TIMPs ex vivo after a preceding orthodontic treatment in vivo; reviews, expert opinions, letters and papers not written in English. No limitations were set concerning used teeth for PDL cell isolation, donor age and gender, force type, mode and frequency and screening methods. Inclusion criteria for in vivo studies were defined as follows: (P) patients undergoing orthodontic treatment; (I) applying orthodontic forces to achieve OTM; (C) teeth not exposed to orthodontic forces and/or samples which were taken before orthodontic forces were applied; (O) evaluation of expression levels and/or enzymatic activities of MMPs and/or TIMPs in the GCF which surrounds teeth exposed to orthodontic forces. In vivo studies were excluded if they met one of the following exclusion criteria: animal studies; studies investigating tooth movement acceleration and/or studies which investigate additional intervention during OTM; articles focusing on diseased patients (e.g., periodontitis and obesity) with orthodontic treatments that had no untreated healthy control group; expert opinions, reviews, letters and studies not written in English. No restrictions were made concerning patient age and gender, location of GCF sampling, necessity of orthodontic treatment, appliance type and the duration of treatment.

### 4.3. Data Synthesis

The included studies were screened for predefined parameters which were summarised and organized in tabularized form. In each table, studies can be identified by their listed study details (first-author name and year of publication). For in vitro and in vivo studies three different table types were created, respectively, with each of them summarising specific data points.

### 4.4. Risk of Bias Assessment

The appraisal of quality and risk of bias was conducted separately of included in vitro and in vivo studies. The risk of bias of in vitro studies was assessed on the basis of the modified guidelines depicted from Samuel et al. [71]. Methodological and reporting qualities were evaluated by nine discrete criteria, respectively. These methodological and reporting criteria are listed in detail in Appendix A. Each individual criterion was rated with yes or no, implying a lower or a higher bias risk, respectively. The risk of bias of relative to in vivo studies was appraised by adapting the Joanna Briggs Institute Critical Appraisal Checklist for analytical cross-sectional study [72]. The quality of studies was assessed by answering seven questions with yes or no, implying a lower or a higher bias risk, respectively. All quality reporting questions are listed in detail in Table 3.

## 5. Conclusions

In conclusion, this systematic review revealed that orthodontic forces have a significant influence on MMPs and TIMPs in the PDL. It is possible that the exact effect of these mechanical forces on different MMP and TIMP types depend on various treatment parameters, such as the appliance type, the force magnitude, treatment duration, and also on tooth localization, the time point of sample collection and the compression versus tension area. Due to a very high variability in the combination of used intervention parameters, it was not possible to verify the influence of a certain combination on specific MMPs/TIMPs within this systematic review by meta-analysis. While increased MMP concentrations at the compression and tension sites are mainly caused by mechanical force-induced MMP expression, the force-induced TIMP expressions seem to be mainly responsible for downregulating MMP activities at the tension site. This mechanism may contribute to the controlled depletion and formation of the PDL’s ECM at the compression and tension zone, respectively, and finally to the highly regulated OTM.

## Figures and Tables

**Figure 1 ijms-22-06967-f001:**
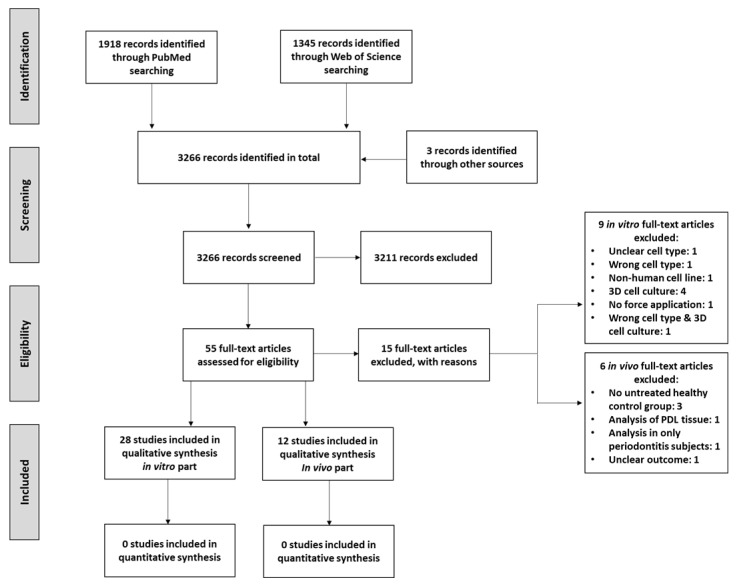
PRISMA Flow Diagram presenting the systematic search results.

**Figure 2 ijms-22-06967-f002:**
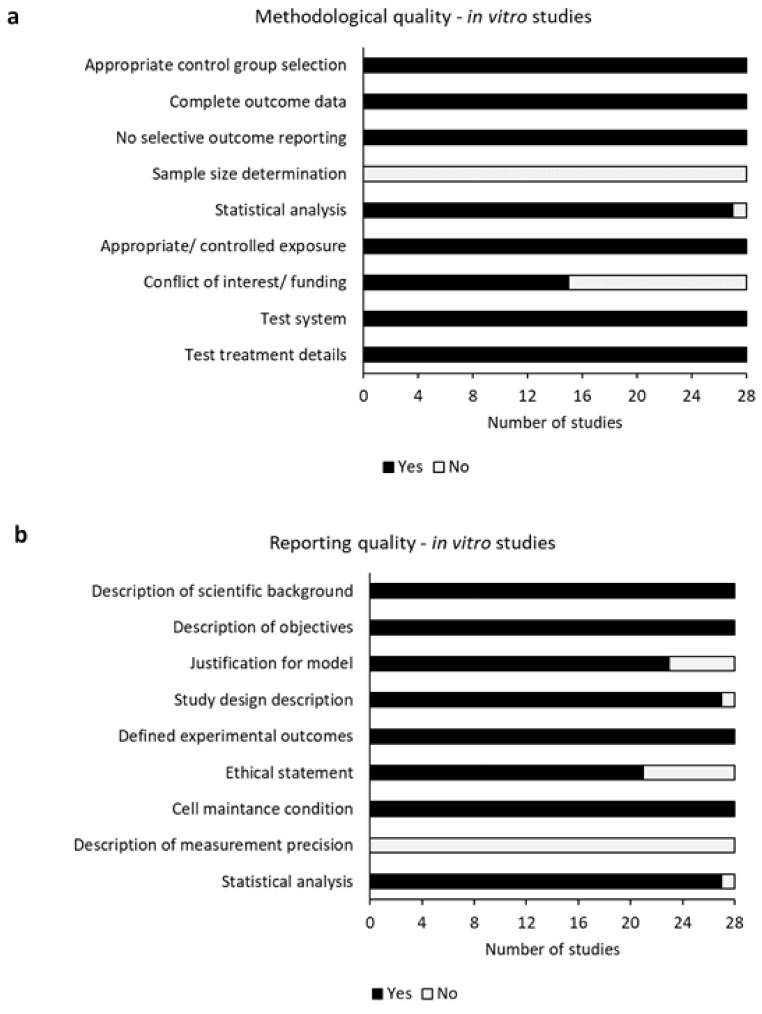
Risk of bias assessment of all included in vitro studies. Risk of bias was evaluated by assessing various methodological (**a**) and reporting (**b**) criteria, which was adapted from Samuel et al., 2016.

**Figure 3 ijms-22-06967-f003:**
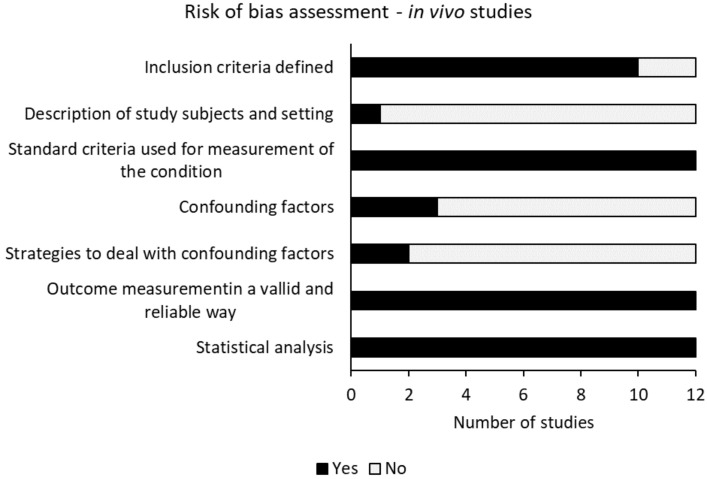
Risk of bias assessment of all included in vivo studies. Risk of bias was evaluated by assessing specific criteria, which was adapted from the Joanna Briggs Institute Critical Appraisal Checklist for analytical cross-sectional study.

**Table 1 ijms-22-06967-t001:** Sample parameter of all included in vitro studies.

Author	Year	Cells	Teeth	Donor Gender	Donor Age	Number of Included Donors
Behm et al.	2021	Human PDL cells (hPDL-MSCs)	Third molars	-	-	4 donors
Bolcato-Bellemin et al.	2000	Human PDL fibroblasts	-	Female	-	1 donor
Chen et al.	2013	Human PDL fibroblasts	Premolars	-	“young”	3 donors
Fujihara et al.	2010	Human PDL cells	-	-	-	-
Grimm et al.	2020	Human PDL fibroblasts	-	-	-	Commercially available HPdLFs
Hacopian et al.	2011	Human PDL fibroblasts	Premolars	-	-	3 donors
Huang et al.	2008	Human PDL cells	Premolars	Female	4.5 ± 1.8 years	16 donors
Jacobs et al.	2014	Human PDL fibroblasts	-	-	-	Commercially available HPdLFs
Jacobs et al.	2018	Human PDL fibroblasts	-	-	-	Commercially available HPdLFs
Kook et al.	2011	Human PDL fibroblasts	-	Male	20–30 years	-
Lisboa et al.	2009	Human PDL fibroblasts	Third molars	Male + female	-	36 donors (43 teeth, only 26 used)
Lisboa et al.	2013	Human PDL fibroblasts	Third molars	-	-	-
Lisboa et al.	2013	Human PDL fibroblasts	Third molars	-	-	-
Long et al.	2002	Human PDL cells	Impacted healthy third molars	Male + female	16 + 18 years	At least 3 donors
Ma et al.	2015	Human PDL cells	Premolars	Female	11 years old	1 donor
Narimiya et al.	2017	Human PDL cell line	-	-	-	Human immortalized PDL cell lines
Nemoto et al.	2010	Human PDL cells	Third molars	-	2 × 20 years, 1 × 40 years	3 donors
Nettelhoff et al.	2016	Human PDL fibroblasts	-	-	-	Commercially available HPdLFs
Proff et al.	2014	Human PDL fibroblasts	-	Male	18–25 years	4 donors
Redlich et al.	2004	Human PDL fibroblasts	Premolars	-	-	At least 3 donors
Saminathan et al.	2012	Human PDL cells	Premolars	-	-	-
Schröder et al.	2020	Human PDL fibroblasts	-	Male + female	17–27 years	6 donors
Tantilertanant et al.	2019	Human PDL cells	-	-	-	3 donors
Tsuji et al.	2004	Human PDL cells	Premolars	-	“young”	-
Wescott et al.	2007	Human PDL fibroblasts	Premolars	-	-	-
Zheng et al.	2012	Human PDL cells	Third molars	-	12–28 years	-
Zheng et al.	2019	Human PDL cells	Third molars	-	12–28 years	>2 donors for each experiment
Ziegler et al.	2010	Human PDL fibroblasts	Premolars	-	12–14 years	-

**Table 2 ijms-22-06967-t002:** Intervention parameters of all included in vitro studies.

Author	Year	Strain Type	Mode	Frequency	Magnitude	Force Duration	Observation Time Points
Behm et al.	2021	Tension	Static	-	6% elongation	6 and 24 h	immediately after force application
Bolcato-Bellemin et al.	2000	Tension	Static	-	20 kPa	12 h	immediately after force application
Chen et al.	2013	Tension	Cyclic	0.1 Hz (6 cycles/min), per cycle: 5 s stretching and 5 s relaxation	3% and 10% elongation	24 and 48 h	immediately after force application
Fujihara et al.	2010	Tension	Cyclic	0.5 Hz (30 cycles/min)	110% elongation	0, 24, and 48 h	immediately after force application
Grimm et al.	2020	Compression	Static	-	34.9 g/cm^2^	3 h	immediately after force application
Hacopian et al.	2011	Compression	Static Cyclic	- 30 min or 15 min with 5 min intervals	36.3 g/cm^2^ 36.3 g/cm^2^	30, 60 and 90 min 90 min	24 h after force application 24 h after force application
Huang et al.	2008	Compression	Static	-	26.5 g/cm^2^; 150 g (180 rpm)	15, 30, 60, or 90 min	immediately after force application
Jacobs et al.	2014	Tension	Static	-	1%, 5% and 10% elongation	12 h	immediately after force application
Jacobs et al.	2018	Tension	Cyclic	-	3% elongation	12 h	immediately after force application
Kook et al.	2011	Tension	Static	-	1.5% elongation	1 h	0–12 h after force application
Lisboa et al.	2009	Compression	Static	-	141 g	30, 60, 90, and 120 min	24, 48, 72 h after force application
Lisboa et al.	2013	Compression	Static	-	141 g	30 min	24 h after stimulation
Lisboa et al.	2013	Compression	Static	-	141 g	30 min	24 h after stimulation
Long et al.	2002	Tension	Cyclic	0.005 Hz	1.8–12.5% elongation	2, 24, 48 h	immediately after force application
Ma et al.	2015	Tension	Cyclic	6 cycles/min, per cycle: 5 s stretch and 5 s relaxation	10% elongation	6 and 24 h	immediately after force application
Narimiya et al.	2017	Tension	Static	-	15% elongation	24 h	immediately after force application
Nemoto et al.	2010	Tension	Cyclic	60 s/returns and 29 s resting time	stretch ratio: 105% stretch length: 1.6 mm	1, 2, 3, 5 and 7 days	immediately after force application
Nettelhoff et al.	2016	Compression	Static	-	2 cN/mm^2^ and 4 cN/mm^2^	12 h	immediately after force application
Proff et al.	2014	Compression	Static	-	2 g/cm^2^	24 h	immediately after force application
Redlich et al.	2004	Compression	Static	-	33.5 g/cm^2^; 167 g	10, 20, 30, 60, 90 and 120 min	immediately after force application
Saminathan et al.	2012	Tension	Cyclic	5 s every 90 s (0.2 Hz)	12% elongation	6, 12 and 24 h	immediately after force application
Schröder et al.	2020	Tension	Static	-	16% elongation	48 h	immediately after force application
Tantilertanant et al.	2019	Tension	Cyclic	60 rpm	10% elongation	2 and 6 h	2, 6, 24 and 48 h
Tsuji et al.	2004	Tension	Cyclic	10 cycles/min, per cycle: 3 s strain and 3 s relaxation	20% elongation	48 h	immediately after force application
Wescott et al.	2007	Tension	Cyclic	6 s strain every 90 s	12% elongation	6, 12, and 24 h	immediately after force application
Zheng et al.	2012	Steady laminar shear flow	Static	-	6, 9 and 12 dyn/cm^2^	2, 4, 8 and 12 h	immediately after force application
Zheng et al.	2019	Steady laminar shear flow	Static	-	6 dyn/cm^2^	4 h	immediately after force application
Ziegler et al.	2010	Tension	Static	-	2.5% elongation	0.25, 0.5, 1, 3, and 6 h	immediately after force application

**Table 3 ijms-22-06967-t003:** MMPs outcome from all included in vitro studies.

Author	Year	Type of Screening	Outcome				
			MMP-1	MMP-2	MMP-3	MMP-8	Other MMPs
Behm et al.	2021	qPCR and ELISA	**=**	**=**			
Bolcato-Bellemin et al.	2000	RT-PCR	**↑** (12 h, 20 kPa)	**↑** (12 h, 20 kPa)			
Chen et al.	2013	RT-PCR, zymogram		**↑** (24 + 48 h, 10%)			
Fujihara et al.	2010	Oligo-DNA chip analysis					MMP-15: **↑** (48 h, 110%)
Grimm et al.	2020	RT-PCR				**↑** (3 h, 34.9 g/cm^2^)	
Hacopian et al.	2011	RT-PCR	**↑** (60 min, 36.3 g/cm^2^)				
Huang et al.	2008	RT-PCR, ELISA	**↑** (changes over time)				
Jacobs et al.	2014	ELISA				**↑** (12 h, 10%)	
Jacobs et al.	2018	ELISA				**↑** (12 h, 3%)	
Kook et al.	2011	RT-PCR	**↑** (1 h, 1.5%)				
Lisboa et al.	2009	Zymogram		**↓** (30–120 min, 141 g)			
Lisboa et al.	2013	ELISA, Western blot analysis	**=**		**=**		MMP-10: **=**
Lisboa et al.	2013	Zymogram		**=**			
Long et al.	2002	RT-PCR, Western blot analysis	**=**		**=**		
Ma et al.	2015	RT-PCR array				**↓** (6 h, 10%)	
Narimiya et al.	2017	RT-PCR, ELISA					MMP-12: **↑** (24 h, 15%)
Nemoto et al.	2010	RT-PCR	**↑** (1–7 d, 105%)		**↓** (1–7 d, 105%)		
Nettelhoff et al.	2016	ELISA				**↑** (12 h, 5 + 10%)	
Proff et al.	2014	RT-PCR					MMP-13: **↑** (24 h, 2 g/cm^2^)
Redlich et al.	2004	RT-PCR	**↑** (30 min, 33.5 g/cm^2^)				
Saminathan et al.	2012	RT-PCR				**↓** (24 h, 12%)	MMP-11: **↓** (12 h, 12%) MMP-15: **↓** (6 h, 12%)
Schröder et al.	2020	RT-PCR				**=**	
Tantilertanant et al.	2019	RT-PCR, ELISA	**↑** (6 h, 10%)	**↑** (6 h, 10%)	**↑** (6 h, 10%)	**=**	MMP-14: **↑** (6 h, 10%)
Tsuji et al.	2004	RT-PCR	**=**	**=**			
Wescott et al.	2007	RT-PCR		**=**		**=**	MMP-9/10: **=**
Zheng et al.	2012	RT-PCR, Western blot analysis	**↑** (4 h, 6–12 dyn/cm^2^)	**↑** (8 h: 12 dyn/cm^2^) (12 h: 6 + 12 dyn/cm^2^)			
Zheng et al.	2019	RT-PCR, Western Blot analysis		**↓** (4 h, 6 dyn/cm^2^)			
Ziegler et al.	2010	RT- PCR, Western blot analysis					MMP-10: **↓** (0.5 h, 2.5%) MMP-12: **↑** (0.5 h, 2–5%) MMP-13: **↑** (0.25 + 0.5 + 6 h, 2.5%)
			8× **↑** 4× **=**	4× **↑** 2× **↓** 4× **=**	1× **↑** 1× **↓** 2× **=**	4× **↑** 2× **↓** 3× **=**	6× **↑** 3× **↓** 3× **=**

**=** unaltered; **↑** increased expression level; **↓** decreased expression level; compared to unstimulated control.

**Table 4 ijms-22-06967-t004:** TIMPs outcome from all included in vitro studies.

Author	Year	Type of Screening	Outcome		
			TIMP-1	TIMP-2	TIMP-3
Behm et al.	2021	RT-PCR	**=**		
Bolcato-Bellemin et al.	2000	RT-PCR	**↑** (12 h, 20 kPa)	**↑** (12 h, 20 kPa)	
Chen et al.	2013	RT-PCR		**↑** (24 h, 10%)	
Fujihara et al.	2010				
Grimm et al.	2020	RT-PCR	**↓** (3 h, 34.9 g/cm^2^)		
Hacopian et al.	2011	RT-PCR	**=**		
Huang et al.	2008				
Jacobs et al.	2014	ELISA	**↑** (12 h, 10%)		
Jacobs et al.	2018	ELISA	**↑** (12 h, 3%)		
Kook et al.	2011				
Lisboa et al.	2009				
Lisboa et al.	2013				
Lisboa et al.	2013				
Long et al.	2002	RT-PCR, Western blot analysis	**=**	**=**	
Ma et al.	2015	RT-PCR array	**↑** (6 + 24 h, 10%)	**=**	**=**
Narimiya et al.	2017	RT-PCR	**=**	**=**	**=**
Nemoto et al.	2010				
Nettelhoff et al.	2016	ELISA	**↓** (12 h, 5 + 10%)		
Proff et al.	2014				
Redlich et al.	2004	RT-PCR	**↑** (10–60 min, 33.5 g/cm^2^)	**↑** (10–60 min, 33.5 g/cm^2^)	
Saminathan et al.	2012	RT-PCR	**=**	**=**	**=**
Schröder et al.	2020				
Tantilertanant et al.	2019	RT-PCR	**=**	**=**	
Tsuji et al.	2004	RT-PCR	**↑** (48 h, 20%)	**↑** (48 h, 20%)	
Wescott et al.	2007				
Zheng et al.	2012	RT-PCR	**↑** (6 dyn/cm^2^: 8 + 12 h (9 dyn/cm^2^: 8 + 12 h) (12 dyn/cm^2^: 4 + 8 + 12 h)	**↓** (6 dyn/cm^2^: 8 + 12 h) (9 dyn/cm^2^: 12 h) (12 dyn/cm^2^: 2 + 8+ 12 h)	
Zheng et al.	2019				
Ziegler et al.	2010				
			7× **↑** 2× **↓** 6× **=**	4× **↑** 1× **↓** 5× **=**	3× **=**

**=** unaltered; **↑** increased expression level; **↓** decreased expression level; compared to unstimulated control.

**Table 5 ijms-22-06967-t005:** Patient parameters of all included in vivo studies.

Author	Year	Gender	Age	Sample Size
Alikhani et al.	2018	-	11–45 years	18
Apajalahti et al.	2003	3 males, 8 females	10-14 and 37-38 years	11
Bildt et al.	2009	2 males, 6 females	10–18 years	8
Cantarella et al.	2006	3 males, 8 females	13–15 years	11
Capelli et al.	2011	3 males, 11 females	12–28 years	14
Grant et al.	2013	-	12–20 years	21
Ingman et al.	2005	2 males, 3 females	11, 12, 13, 13 and 36 years	5
Ribagin et al.	2012	6 males, 6 females	11–15 years	12
Rody et al.	2014	3 males, 7 females 4 males, 6 females	13–15 years 21–39 years	10 adolescents 10 adults
Shirozaki et al.	2020	-	11–44 years	28
Surlin et al.	2014	6 males, 10 females	13–17 years	16
Zhang et al.	2020	-	12–18 years	20

**Table 6 ijms-22-06967-t006:** Intervention parameters of all included in vivo studies.

Author	Year	Teeth	Sample	Appliance Type	Force	Duration	Observation Time	Control
Alikhani et al.	2018	canines	GCF	nickel-titanium closing-coil spring; sequential archwires from 0.016-in nickel-titanium to 0.017 × 0.025-in stainless steel	50 cN	56 days	before, 1, 7, 14, and 28 days after the canine retraction	sample collection immediately before canine retraction
Apajalahti et al.	2003	upper incisor, upper canine or lower central incisor	GCF	fixed appliance	-	-	before OTM and every hour for 8 h following application	systemically healthy patients without OTM
Bildt et al.	2009	mostly upper first premolars	GCF	super-elastic nickel-titanium coil springs	150 cN	4 weeks	4 weeks after starting force application	teeth without appliances
Cantarella et al.	2006	Left upper canines	GCF	0.016-in circular cross-sectional dimension, nickel-titanium orthodontic wire, nickel-titanium coil spring	150 g	-	1, 2, 3, 4, 8 h	antagonistic tooth with no appliance
Capelli et al.	2011	upper canines	GCF	-	150 g	-	7 days before OTM, day of OTM, 1 h, 24 h, 2 weeks, 3 weeks and 80 days after application	sample collection 7 days before orthodontic force applied
Grant et al.	2013	upper canines	GCF	MBT prescription brackets and elastomeric modules, archwire sequence 0.014 nickel-titanium → 0.018 nickel-titanium → 0.018 stainless steel; 9mm nickel-titanium closing coil spring	100 g	-	4 h, 7 days and 42 days	sample collection from test teeth before orthodontic force applied; untreated second molars
Ingman et al.	2005	upper or lower central incisor or upper canine	GCF	fixed appliance treatment (mini-mat brackets, 0.018-inch slot)	-	1 month	just before appliance and then every 24 h	one upper central incisor from each of three healthy females (mean age 36 years) without OTM
Ribagin et al.	2012	first molar (central incisor or canines)	GCF	-	-	>3 months	before OTM (up to 1 week), 24 h after placement, 1 week after last visit, 3 months after placement	sample collection before OTM
Rody et al.	2014	upper arch (upper incisors)	GCF	conventional fixed edgewise bracket system, 0.014-in, 0.018-in, 0.01630.022-in and 0.01930.025-in nickel-titanium	-	20 weeks	Immediately before bonding and after 3, 6, 18 and 20 weeks	mandibular incisors, free from any orthodontic appliance
Shirozaki et al.	2020	upper and lower first molars and upper and lower left central incisors	GCF	brackets (0.022″ × 0.028″ slot of stainless steel with stainless steel wires (0.016″, 0.018″, 0.020″, or 0.019″ × 0.025″) and 4 bands in the first molars	-	-	before, 6 and 12 months after bracket bonding	sample collection before treatment
Surlin et al.	2014	upper canines	GCF	brackets Roth 0.018 inch with 0.012-inch nickel-titanium archwire and a laceback made from 0.010-inch stainless wire	-	until completion of OTM	1 h before application, 1 h, 4 h, 8 h, 24 h, 72 h, 1 and 2 weeks after force application	contralateral canines (no force applied)
Zhang et al.	2020	canines	GCF	-	-	-	day of application, 1 h, 24 h, 1 week, 4 weeks and 12 weeks after force application	contralateral teeth of same arch without orthodontic treatment and sample collection before treatment

**Table 7 ijms-22-06967-t007:** MMPs outcome of all included in vivo studies.

Author	Year	Type of Screening	Outcome						
			MMP-1	MMP-2	MMP-3	MMP-8	MMP-9	MMP-13	MMP-14
Alikhani et al.	2018	Glass slide-based protein assay					**↑** (after 1 + 7 + 14 days)		
Apajalahti et al.	2003	Immunofluorometric assay, western blot analysis	not detectable			**↑** (after 4–8 h)			
Bildt et al.	2009	Zymogram, western blot analysis	compression/tension: **↑** (after 4 weeks)	compression/tension: **↑** (after 4 weeks)		not detectable	compression: **↑** (after 4 weeks)	not detectable	
Cantarella et al.	2006	Western blot analysis	compression: **↑** (after 1 + 2 + 3 h) tension: **↑** (after 1 + 2 h)	compression: **↑** (after 1 + 2 + 3 + 4 + 8 h) tension: **↑** (after 1 + 2 + 3 + 4 h)					
Capelli et al.	2011	Multiplexed bead immunoassay			compression: changes over time		compression: changes over time	compression: changes over time	
Grant et al.	2013	Luminex multi-analyte technology					compression/tension: **↑** (changes over time)		
Ingman et al.	2005	Immunofluorometric assay, western blot analysis	not detectable			**↑**			
Ribagin et al.	2012	ELISA				**↓** (1 week) **↑** (after 3 months)			
Rody et al.	2014	Microarray assay					**=**		
Shirozaki et al.	2020	Milliplex TM Map, multiplexing analyser MAGPIX				**=**			
Surlin et al.	2014	ELISA					**↑** (after 4 + 8 h and 1 + 2 weeks)		
Zhang et al.	2020	Multiplex Luminex, Taqman microRNA assays	compression/tension: **↑** (after 24 h + 1 + 4 weeks)	compression/tension: **↑** (after 24 h + 1 + 4 weeks)	compression/tension: **↑** (after 24 h + 1 + 4 weeks)	compression/tension: **↑** (after 24 h + 1 + 4 weeks)	compression/tension: **↑** (after 24 h + 1 + 4 weeks)	compression/tension: **↑** (after 24 h + 1 + 4 weeks)	compression/tension: **↑** (after 24 h + 1 + 4 weeks)
			3× **↑** 2× not detectable	3× **↑**	1× **↑**	4× **↑** 1× **↓** 1× **=** 1× not detectable	5× **↑** 1× **=**	1× **↑** 1× not detectable	1× **↑**

**=** unaltered; **↑** increased expression level; **↓** decreased expression level.

**Table 8 ijms-22-06967-t008:** TIMPs outcome of all included in vivo studies.

Author	Year	Type of Screening	Outcome	
			TIMP-1	TIMP-2
Alikhani et al.	2018			
Apajalahti et al.	2003			
Bildt et al.	2009	Reverse zymogram	compression/tension: **↑** (after 4 weeks)	not detectable
Cantarella et al.	2006			
Capelli et al.	2011			
Grant et al.	2013	Luminex multi-analyte technology	compression/tension: **↑** (changes over time)	compression/tension: **↑** (changes over time)
Ingman et al.	2005			
Ribagin et al.	2012			
Rody et al.	2014			
Shirozaki et al.	2020			
Surlin et al.	2014			
Zhang et al.	2020			
			2× **↑**	1× **↑** 1× not detectable

**↑** increased expression level; **↓** decreased expression level.

## Data Availability

The data used to support the findings of this study are available from the corresponding author upon request. The data supporting this systematic review are from previously reported studies and datasets, which have been cited.

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
