# Peer review of "MMPs and TIMPs Expression Levels in the Periodontal Ligament during Orthodontic Tooth Movement: A Systematic Review of In Vitro and In Vivo Studies"

_ijms, 2021, doi:10.3390/ijms22136967_

Round 1
Reviewer 1 Report
At the end of the discussion section, it would be useful to present a suggested protocol for this kind of research, based on the analysis of the existing literature performed in this review.
Why only PubMed was searched? Other databases, such as for example Scopus and Web of Science, should be searched. Language should not be restricted to English only.
GCF: the meaning of this abbreviation must be reported the first time it is used
Line 680. “It possible that the exact” must be “It is possible that the exact”
Supplementary files are missing: the link in the manuscript is not working.
Author Response
Reviewer 1
Comment 1
At the end of the discussion section, it would be useful to present a suggested protocol for this kind of research, based on the analysis of the existing literature performed in this review.
Authors’ answer
We thank reviewer 1 for this interesting suggestion. Within the discussion section we added some recommendations which points have to be considered in future studies in this research field. Concerning in vitro studies an important point is the high heterogeneity of used parameter combinations and cell donor variability. To overcome this issue, we recommend in the discussion section to test and directly compared MMPs expression levels under different combinations of force type, application mode and force magnitude and duration. Further, the usage of cells from the same passage and donor will reduce donor variability.
Concerning in vivo studies, we recommend to precisely describe control groups and substantiate the choice of the control type, since investigated studies show a heterogeneity in the choice of the control type. Additionally, we recommend to investigate MMPs and TIMPs expression levels directly in the periodontal ligament.
Comment 2
Why only PubMed was searched? Other databases, such as for example Scopus and Web of Science, should be searched. Language should not be restricted to English only.
Authors’ answer
We thank the reviewer for this important comment. Due to the comment of the reviewer, we have also searched Web of Science. We have adopted the PubMed search strategy to Web of Science and found 1345 papers. After screening the Web of Science search according to the PICO criteria and our defined inclusion and exclusion criteria, no additional paper could be included into this systematic review. All this information were added into the Material and Methods sections and the results sections.
In our opinion, non-English written papers have to be excluded from this systematic review, since data extraction and performing “Risk of Bias Assessment” would not be possible. We added the exclusion of non-English written papers as a limitation of this systematic review in the discussion section.
Comment 3
GCF: the meaning of this abbreviation must be reported the first time it is used
Authors’ answer
We have explained this abbreviation at its first appearance in the results section.
Comment 4
Line 680. “It possible that the exact” must be “It is possible that the exact”
Authors’ answer
We have revised this sentence.
Comment 5
Supplementary files are missing: the link in the manuscript is not working.
Authors’ answer
We apologize for this. As far as we know, the journal provides supplementary files to the Reviewers without any link in the manuscript draft. We have uploaded supplementary files together with the main manuscript.
Reviewer 2 Report
Journal: International Journal of Molecular Sciences
Manuscript ID: ijms-1257529
Type of manuscript: Systematic Review
Title: MMPs and TIMPs expression levels in the periodontal ligament during orthodontic tooth movement: a systematic review of in vitro and in vivo studies
Authors: Christian Behm, Michael Nemec, Fabian Weissinger, Xiaohui Rausch-Fan, Oleh Andrukhov *, Erwin Jonke
Submitted to section: Macromolecules
General comments:
In the present review, which was performed according to the systematic reference search, the effects of specific orthodontic forces on the expressions of matrix metalloproteinases (MMPs) and their inhibitors tissue inhibitor of matrix metalloproteinases (TIMPs) in periodontal ligament (PDL) were investigated. The objective was to characterize the expression levels of MMPs and TIMPs during orthodontic tooth movement after the application of specific orthodontic forces. Studies conducted in vitro (28 studies) and in vivo (12 studies) were selected and analyzed according to objective parameters. Two main outcomes were found: (1) Given the very high variability in the combination of treatment parameters used, the authors could not determine to characterize the effects of a particular combination on specific MMPs/TIMPs in PDL. (2) The mechanical forces induce increased expressions of MMP at the compression and tension sites. Increased expressions of TIMP by the mechanical forces occur at the compression and tension sites where the expressions of MMP decrease. From these results, the authors concluded that the expression levels of MMP and TIMP in PDL could contribute to the controlled degradation and formation of extracellular matrix in the compression and tension zones during orthodontic tooth movement.
This is an interesting review. The analyzed studies with the controls were well selected and the results were well discussed.
Minor point:
Some references should be corrected according to Journal guidelines.

Author Response
Reviewer 2
Comment 1
In the present review, which was performed according to the systematic reference search, the effects of specific orthodontic forces on the expressions of matrix metalloproteinases (MMPs) and their inhibitors tissue inhibitor of matrix metalloproteinases (TIMPs) in periodontal ligament (PDL) were investigated. The objective was to characterize the expression levels of MMPs and TIMPs during orthodontic tooth movement after the application of specific orthodontic forces. Studies conducted in vitro (28 studies) and in vivo (12 studies) were selected and analyzed according to objective parameters. Two main outcomes were found: (1) Given the very high variability in the combination of treatment parameters used, the authors could not determine to characterize the effects of a particular combination on specific MMPs/TIMPs in PDL. (2) The mechanical forces induce increased expressions of MMP at the compression and tension sites. Increased expressions of TIMP by the mechanical forces occur at the compression and tension sites where the expressions of MMP decrease. From these results, the authors concluded that the expression levels of MMP and TIMP in PDL could contribute to the controlled degradation and formation of extracellular matrix in the compression and tension zones during orthodontic tooth movement.
This is an interesting review. The analyzed studies with the controls were well selected and the results were well discussed.
Authors’ answer
We thank reviewer 2 for this overall positive feedback.
Comment 2
Minor point:
Some references should be corrected according to Journal guidelines.
Authors’ answer
We apologize for this mistake and thank the reviewer for this criticism. We have carefully revised the reference list in accordance to the journals’ reference guidelines.
Round 2
Reviewer 1 Report
In this revised version of the manuscript, all my previous comments were addressed
Author Response
Thank you for your positive feedback.